# Nucleation Process in Explosive Boiling Phenomena of Water on Micro-Platinum Wire

**DOI:** 10.3390/e26010035

**Published:** 2023-12-28

**Authors:** Yungpil Yoo, Ho-Young Kwak

**Affiliations:** 1Department of Climate Change Energy Engineering, Yonsei University, Seoul 03722, Republic of Korea; yyp@besico.co.kr; 2Blue Economy Strategy Institute Co., Ltd., #602, 150 Dogok-ro, Gangnam-gu, Seoul 06260, Republic of Korea; 3Mechanical Engineering Department, Chung-Ang University, Seoul 06974, Republic of Korea

**Keywords:** superheated limit, molecular interaction model, explosive boiling, bubble nucleation, micro-platinum wire, water

## Abstract

The maximum temperature limit at which liquid boils explosively is referred to as the superheat limit of liquid. Through various experimental studies on the superheating limit of liquids, rapid evaporation of liquids has been observed at the superheating limit. This study explored the water nucleation process at the superheat limit achieved in micro-platinum wires using a molecular interaction model. According to the molecular interaction model, the nucleation rate and time delay at 576.2 K are approximately 2.1 × 10^11^/(μm^3^μs) and 5.7 ns, respectively. With an evaporation rate (116.0 m/s) much faster than that of hydrocarbons (14.0 m/s), these readings show that explosive boiling or rapid phase transition from liquid to vapor can occur at the superheat limit of water. Subsequent bubble growth after bubble nucleation was also considered.

## 1. Introduction

Typically, boiling of solid surfaces occurs at low superheat temperatures, between 1 K and 10 K, and is called heterogeneous nucleation or nucleate boiling [1]. Clark et al. [2] identified nucleate boiling sites as pits on the surface using high-speed cameras. Cornwell [3] identified natural sites where nucleate boiling occurred on the copper surface via scanning electron microscopy (SEM). Nail et al. [4] identified the nucleate boiling sites via SEM in a cavity with a radius ranging from 0.4 to 5 μm. Nucleate boiling is characterized by bubble departure frequency (ms) and the number of nucleation sites, which are related to the boiling heat transfer [5]. Trefethen [6] showed that a liquid can overheat when it is surrounded by another liquid. Inspired by this observation, Wakeshima and Takata [7] and Moore [8] performed droplet explosion experiments to obtain the theoretical superheat limit [9] at which homogeneous nucleation occurs. Since then, atomic-scale smooth surfaces without cavities have been fabricated using integrated circuit (IC)-based microfabrication techniques. On microscale, atomic-scale smooth surfaces, boiling occurs at very large superheats near the critical temperature of the liquid and proceeds rapidly, on the order of microseconds [10].

The maximum temperature limit at which liquid boils explosively is referred to as the superheat limit of liquid, which can be achieved using various techniques. In the “droplet explosion technique”, a droplet of a less-dense test liquid is heated by suspending it in a test tube containing an immiscible medium with a vertical temperature gradient [7,11] to determine its superheat limit. The superheat limit of hydrocarbons measured by the droplet explosion techniques is approximately 89–90% of the critical temperature [12,13,14,15]. The droplet explosion technique yielded a superheat limit of 468 K for methanol. Eberhart et al. [16] reported a superheat limit of 459 K by heating the liquid in a capillary tube. When a liquid droplet in the hot host liquid reaches its superheat limit, it evaporates explosively [17]. This rapid phase transition from liquid to vapor is called explosive boiling. A 1 mm diameter butane droplet completely evaporates at the superheat limit in ethylene glycol in approximately 40–60 μs, implying that the evaporation rate at the superheat limit is greater than 14 m/s [17,18]. However, the maximum superheat limit of water obtained using the droplet explosion technique in benzyl benzoate was approximately 552.7 K [19], which is considerably lower than what is considered the superheat limit of water.

The classical bubble nucleation theory predicts the superheat limit for hydrocarbons by assuming a nucleation rate of 10^6^ bubbles/(cm^3^s) [11]. However, the molecular interaction model predicts the superheating limit for butane by assuming a nucleation rate of 10^22^ clusters/(cm^3^s) or 10^13^ clusters/(mm^3^μs) [20]. The molecular cluster model’s high nucleation rate predicts the evaporation of 1 mm diameter droplets in tens of microseconds at the superheat limit of butane. However, the nucleation process at the superheat limit, which occurs in tens of microseconds within a cubic mm volume, cannot be explained by classical bubble nucleation theory. The superheat limit of a liquid and the droplet evaporation duration at that limit can be predicted using nucleation theory.

The superheat limit of a liquid can be obtained using various experimental methods and procedures. A liquid can be heated using high-power laser irradiation [21,22]. When a liquid absorbs photons from a laser pulse, the liquid volume around the laser focus can reach or exceed its superheat limit [23]. Byun and Kwak [24] calculated the temperature and initial wall velocity as 576.1 K and 620 m/s, respectively, of a laser-induced bubble that emits optical emission on collapsing and has a maximum radius of 1.05 mm and a half-width of 11.4 ns. Water rapidly changed its phase from liquid to vapor at 603 K near the core of the membrane nanopore when a voltage was applied to the two electrodes on either side of a thin silicon nitride membrane in a 3 M NaCl solution [25].

The superheat limit of water was reached by pulse heating an ultra-thin platinum (Pt) wire [26,27,28]. Derewnicki obtained a water superheat limit of 573.2 K in a platinum (Pt) wire with a 25 μm diameter at 1 atm. Skripov and Pavlov [26] obtained a water superheat limit of 575.2 K using the pulse heating method. Glod et al. [28] heated a 10 μm diameter Pt wire at a heating rate of 60 × 10^6^ K/s to obtain a maximum water superheat of 576.2 K. They observed that the Pt surface was covered with a thin vapor film immediately after heating at high heating rates.

By pulsing current to the corresponding heaters, bubble nucleation can be observed in thin film microheaters [29,30], microheaters in bubble jet printers [31], and micro-line heaters [10] fabricated using standard IC processes [32]. Iida et al. [30] obtained a water superheat limit of 578.2K by heating at a heating rate of 3.77 × 10^7^ K/s in a 100 × 250 μm microheater. Avedisian et al. [33] reported that the bubble nucleation temperature (about 556 K) measured in a Ta/Al inkjet printer heater with dimensions of 65 × 65 μm may be predicted by a molecular cluster model with a surface nucleation rate of 10^13^ nuclei/(cm^2^s). However, at the same nucleation rate, the bubble nucleation temperature at the heater surface was predicted to be 586 K using the classical bubble nucleation theory. The boiling temperature on the micro-line heater with dimensions of 2 × 50 μm [10] or 3 × 50 μm [10,34] was close to or exceeded the superheat limit of Fluorinert liquids. Bubble formation and growth were controlled by the amount and duration of the input current to the heater [34]. The superheat limits of methanol, ethanol, butanol, and n-heptane were obtained by pulse heating a stress-minimized platinum film heater supported by a SiN membrane at a rate above 10^8^ K/s [35]. The smallest reduced temperature measured in their experiments was in the range of 0.83–0.85, ruling out the possibility of heterogeneous nucleation. The heater sizes used in their experiment were 4 × 60 μm and 4 × 80 μm. Kozulin and Kuzntsov [36] observed the explosive boiling of water, propanol-1, propanol-2, and n-nonane at heating rates of 1–4 × 10^8^ K/s in a multilayer thin film resistor of 100 × 110 μm. The resistors used in their experiments were a four-layer film deposited layer-by-layer using the plasma-enhanced chemical deposition (PECVD) method on a thin glass substrate. The microheater roughness obtained by atomic force microscopy with a resolution of 5 × 5 μm was approximately 1.54 nm. However, even if the heater surface is smooth at the atomic level, typical nucleate boiling occurs in a large heater with dimensions of 2.3 × 2.3 cm [37].

Bubble formation on the surface of gold nanoparticles [38,39,40] or microparticles [41] heated with a high-power laser was observed experimentally and theoretically. When the liquid layer is heated by thermal diffusion from the heater surface of the Pt wire and the silicon wafer or gold particle surface reaches the superheat limit, a phase transition from liquid to vapor may occur. Molecular dynamics simulation studies [42] showed that the water layer closest to the heated metal surface overheated and exploded, and the vaporized molecules exerted forces on other water layers.

A liquid evaporates at its superheat limit, which can be achieved using the aforementioned methods and techniques. Herein, the molecular interaction model for bubble nucleation was used to study the nucleation process, which can be described by the evaporation of the water layer in an ultrathin Pt wire. The nucleation rate of the critical cluster, the number of molecules within the critical cluster, the evaporation rate of the liquid layer, and the time delay for nucleation were obtained at the nucleation temperature of water measured on the Pt wire. Based on the classical nucleation theory, the nucleation rate at the superheat limit of water on ultrathin Pt wires (576.2 K) was approximately 3.3 × 10^−14^ bubbles/(μm^3^μs), indicating that nucleation did not occur. However, when the heating rate for the wire exceeded 74~86 × 10^6^ K/s [28], the molecular interaction model for bubble nucleation showed that bubble nucleation at the water superheat limit on the Pt wires produced a nucleation rate of 2.2 × 10^11^ clusters/(μm^3^μs). This indicated the occurrence of explosive boiling.

## 2. Bubble Nucleation Theory and Model

### 2.1. Classical Theory of Bubble Nucleation

The main assumption of classical bubble nucleation theory is that a critical-size bubble in mechanical equilibrium is formed. The critical-size bubble for overcoming the macroscopic surface tension can be expressed as
(1)$$P_{e}-P_{f}=\frac{2\sigma }{R_{c}}$$
where *R_c_* is the radius of the critical-size bubble and *σ* is the surface tension measured. The equilibrium pressure of the bubble *P_e_* in solution is related to the initial saturation pressure, *P_sat_,* at a given liquid temperature *T*, and the ambient pressure, *P_f_* [43]. That is
(2)$$P_{e}=P_{sat}\exp\left(-\frac{V_{m}(P_{sat}-P_{f})}{k_{B}T}\right)$$
where *V_m_* and *k_B_* are the molecular volume of liquid and Boltzmann constant. The number of molecules constituting the critical-size bubble can be calculated using the ideal gas law.

The maximum energy required to form a critical-size bubble, *F_Rc,_* as determined by a thermodynamic argument based on availability analysis [1], is given by
(3)$$F_{R_{c}}=\frac{4\pi R_{c}^{2}\sigma }{3}=\frac{16\pi \sigma^{3}}{3{\left(P_{e}-P_{f}\right)}^{2}}$$Equations (2) and (3) proposed by Gibbs [44] have been widely used previously to predict bubble formation in solutions [11,45].

Doering [9], Volmer and Weber [46], and Zeldovich [47] developed the kinetics for the classical nucleation theory, which provides steady-state processes of nucleation mechanisms for the formation of the critical droplet. The bubble nucleation rate can be calculated using the kinetic theory for the droplet case, assuming ideal behavior for the gas inside the critical-size bubble. A detailed derivation is given in Debendetti [48]:(4)$$J_{CNT}=N{\left(\frac{3\sigma }{2\pi m}\right)}^{1/2}\cdot \exp\left[-\frac{16\pi \sigma^{3}}{3k_{B}T{\left(P_{e}-P_{f}\right)}^{2}}\right]$$
where *N* is the number density of a liquid at a given temperature and *m* is the molecular mass of a liquid molecule.

Equation (4) can be used to estimate the superheat limit of the liquid using assumed or measured nucleation rate values. Equation (4), as is well known, predicts the superheat limits for hydrocarbons, halocarbons, and several alcohols, assuming a nucleation rate value of 10^6^ bubbles/(cm^3^s) [11]. However, Equation (4) lacks information regarding the evaporation process of the liquid at the superheating limit, which has been confirmed by experimental and theoretical studies of the superheat limit of liquids. Snitsyn and Skripov [49] established the time scale of the nucleation event t_s_ to estimate the lifetime of the droplet at the superheat limit as:(5)$$t_{s}=1/(J_{CNT}V_{d})$$
where *V_d_* is the droplet volume. Equation (5) is the time scale of the oscillation of the bubble formed after the droplet explosion is complete, in milliseconds using a droplet diameter of 1 mm and a nucleation rate of 10^6^ bubbles/(cm^3^s), which are typical sizes used in droplet explosion experiments.

### 2.2. Molecular Interaction Model for Vapor Bubble Nucleation

A clustering process of activated molecules is thought to occur in metastable liquids. Bubble formation in a liquid is analogous to separating the intermolecular distances of molecules within a cluster to a critical state distance. This is because the fluid at the critical point has no surface tension and can thus be considered a gas. Assuming that the intermolecular interactions in a metastable liquid are van der Waals, the energy required to separate a pair of molecules from their average distance in the liquid state to their average distance at the critical point is [50]
(6)$$\varepsilon_{m}=4\varepsilon_{o}\left[1-{\left(\frac{\rho_{c}}{\rho_{m}}\right)}^{2}\right]\left[{\left(\frac{d_{w}}{d_{m}}\right)}^{6}-{\left(\frac{d_{w}}{d_{m}}\right)}^{12}\right]$$
where *d_w_* is the van der Waals diameter [51], *d_m_* is the average distance between molecules at a given temperature, $\rho_{c}$ and $\rho_{m}$ are the density at the critical point and current state, respectively. The potential of the van der Waals interaction *ε_o_* is given by [52]
(7)$$\varepsilon_{o}=\frac{3}{16}\frac{E_{I}\alpha^{2}}{d_{w}^{6}}$$
where *E_I_* is the ionization potential and α is the polarizability of the molecule.

If a single molecule is surrounded by its nearest *Z* neighbors (*Z* = 12 for the face-centered cubic structure), the energy *W* = *Zε_m_*/2 [53] can be used to detach the molecule from the group, where *Z* is the number of nearest neighbor molecules. The energy, *W_n_*, required to cut across a cluster of n molecules from the surrounding liquid [54] is as follows:(8)$$W_{n}=W\cdot n^{2/3}=\frac{Z}{2}\varepsilon_{m}n^{2/3}$$

When a saturated liquid is depressurized, a metastable state develops, and the difference in chemical potential between the metastable and the saturated states provides the driving force for cluster formation in the metastable state, as indicated by
(9)$$\mu_{m}-\mu_{s}=V_{m}(P_{f}-P_{v})$$
where $\mu_{m}$ and $\mu_{s}$ are the chemical potential of metastable and saturated states, respectively, and *P_v_* is the vapor pressure given the temperature. Then the free energy associated with cluster formation in a metastable state is given by [50].
(10)$$F_{n}=(P_{f}-P_{v})nV_{m}+\frac{Z}{2}\varepsilon_{m}n^{2/3}$$The condition in which the critical cluster is formed can be obtained by maximizing *F_n_* with respect to *n* in Equation (10). The stability condition of the cluster is given as
(11)$$P_{v}-P_{f}=\frac{1}{n_{c}^{1/3}}\;\left(\frac{Z\varepsilon_{m}}{3V_{m}}\right)$$Substituting Equation (11) into Equation (10), the maximum free energy for the critical cluster formation is given as
(12)$$F_{n_{c}}=\frac{Z}{6}\left(\frac{\varepsilon_{m}}{k_{B}T}\right)\;n_{c}^{2/3}$$

The steady state nucleation rate of the critical clusters can be expressed using the kinetic theory argument [55]:(13)$$J_{vol}=ND_{n}Z_{f}\exp\left[-\frac{Z}{6}\frac{\varepsilon_{m}}{k_{B}T}\;n_{c}^{2/3}\;\right]$$
where *Z_f_* is the Zeldovich nonequilibrium factor [56], which can be obtained from the free energy of the critical cluster. That is
(14)$$Z_{f}={\left[-\frac{1}{2\pi k_{B}T}{\left(\frac{\partial^{2}F_{n}}{\partial^{2}n}\right)}_{{n=n}_{c}}\;\right]}^{1/2}={\left(\frac{Z\varepsilon_{m}}{54\pi k_{B}T}\right)}^{1/2}n_{c}^{-2/3}$$The unit of nucleation rate *J_vol_*, expressed for volume nucleation, is clusters/(cm^3^s). *D_n_* is the striking rate of molecules on the surface of the cluster with n molecules, which is given as
(15)$$D_{n}=\frac{\beta }{4}N\left(\frac{8k_{B}T}{\pi m}\right)\exp\left[-\frac{\Delta H_{vap}}{RT}-\frac{\Delta H_{f}}{RT_{f}}\right]4\pi {\left(\frac{3V_{m}}{4\pi }\right)}^{2/3}n^{2/3}$$
where *β* is the accommodation coefficient, which was taken in this study. The time lag, which represents the duration of the transient state after the onset of nucleation, is given by [56]
(16)$$t_{l}=1/{\left(4\pi D_{n}Z_{f}^{2}\right)}_{n=n_{c}}$$Using Equation (13), we can calculate the number of molecules within the critical cluster given the nucleation rate of the critical cluster. The pressure of the cluster can be calculated using Equation (11), but the pressure of the evaporated state with the liquid volume retained is large, as seen in Equation (17) [20]:(17)$$P_{n}=\frac{Z\varepsilon_{m}}{3V_{m}}$$The pressure of the evaporated water is approximately 827 bar at the superheat limit of 566.2 K, causing its volume to expand.

## 3. Nucleation on Atomic-Scale Smooth Surfaces

Surface nucleation, which is discussed in this section, can be considered when boiling water on micro-Pt wires. Consider a bubble formed on an atomic-scale smooth surface with a liquid-side contact angle of *θ*, as shown in Figure 1. The liquid–gas and solid–gas interface areas, *A_lg_*, and *A_sg_*, and the volume of the bubble, *V_b_,* are given as [11]
(18)$$A_{\mathrm{lg}}=4\pi R^{2}(1+\cos\theta )/2=4\pi R^{2}\Psi$$
(19)$$A_{sg}=4\pi R^{2}(1-\cos^{2}\theta )/4=4\pi R^{2}\Theta$$
(20)$$V_{b}=\frac{4\pi R^{3}}{3}(2+3\cos\theta -\cos\theta^{3})/4=\frac{4\pi R^{3}}{3}\Phi$$
where Ψ, Θ, and Φ are the correction factors for the liquid–gas and solid–gas surfaces and volume of the bubble, respectively, whereas *R* is the radius of curvature of the bubble.

As in classical nucleation theory for surface nucleation [11], the free energy of the formation of clusters can be obtained by applying the surface and volume correction factors to the driving force term and the surface energy term for the free energy in the molecular interaction model.
(21)$$F_{n}=-(P_{i}-P_{f})nV_{m}\Phi +\frac{Z}{2}\varepsilon_{m}\cdot n^{2/3}\;\left[\Psi +\cos\theta \cdot \Theta \right]$$
where Ψ + cos*θ*Θ = Φ. In the derivation of Equation (21), the following relation between the interfacial tension is used:(22)$$\sigma_{lg}A_{lg}+\left[\sigma_{sg}-\sigma_{sl}\right]A_{sg}=\sigma_{lg}\cdot \;4\pi R^{2}\;\left[\Psi +\cos\theta \cdot \Theta \right]$$The last term in Equation (22) indicates that the molecules at the gas–solid interface are excluded. Then, the free energy for the formation of the *n_c_*-mer cluster changes, such that
(23)$$F_{n_{c}}=\frac{Z}{6}\left(\frac{\varepsilon_{m}}{k_{B}T}\right)n_{c}^{2/3}\Phi$$

When impinging molecules are considered at the liquid–gas interface, the nucleation rate (clusters/cm^2^s) of the critical clusters on the atomic-scale smooth surface is given as
(24)$$\begin{array}{cc} J_{sur} & =N{\left(\frac{k_{B}T}{2\pi m}\right)}^{1/2}\left[\frac{Z}{18\pi }\left(\frac{\varepsilon_{m}}{k_{B}T}\right)\right]4\pi {\left(\frac{3V_{m}}{4\pi }\right)}^{2/3}\Psi \\ & \times \exp\left[-\frac{\Delta H_{vap}}{R_{g}T}-\frac{\Delta H_{f}}{R_{g}T_{f}}\right]N^{2/3}/\sqrt{\Phi }\exp\left[-\frac{Z}{6}\left(\frac{\varepsilon_{m}}{k_{B}T}\right)n_{c}^{2/3}\Phi \right] \end{array}$$The unit of nucleation rate for surface nucleation *J_sur_* is clusters/(cm^2^s). The free energy and nucleation rate for bubble nucleation on smooth surfaces at the atomic scale can be applied to bubble formation on flat surfaces with limited movement of molecules in one hemisphere [33]. Equations (23) and (24) can be applied to the boiling of hydrophobic surfaces with large contact angles with liquid [57].

The nucleation rate of classical nucleation theory for surface nucleation is given by [11]
(25)$$J_{CNT,sur}=N^{2/3}\Psi {\left(\frac{2\sigma }{\pi m\Phi }\right)}^{1/2}\cdot \exp\left[-\frac{16\pi \sigma^{3}\cdot \Phi }{3k_{B}T{\left(P_{e}-P_{f}\right)}^{2}}\right]$$

## 4. Bubble Dynamics

Kwak et al. [58] studied the oscillation of bubbles from fully evaporated liquid droplets at the superheat limit analytically. The following is a summary of this study’s application to the behavior of bubbles formed in micro-Pt wires. The well-known Rayleigh–Plesset equation, which can be obtained from the mass and momentum equations of an incompressible medium by adding surface tension and dynamic viscosity factors, can be used to describe bubble wall motion [59]:(26)$$R_{b}\frac{dU_{b}}{dt}+\frac{3U_{b}^{2}}{2}=\frac{1}{\rho_{\infty }}\left(P_{b}-P_{\infty }-\frac{2\sigma }{R_{b}}-\frac{4\mu U_{b}}{R_{b}}\right)$$
where *R_b_* and *U_b_* are the instantaneous bubble wall radius and velocity, respectively, *P_b_* is the pressure inside the bubble, and μ is the dynamic viscosity of the liquid. Assuming that the gas inside the bubble obeys the ideal gas law and has a spatially uniform temperature, the time-dependent pressure inside the bubble may be obtained using overall energy conservation, including heat transfer through the bubble walls:(27)$$\frac{dP_{b}}{dt}=-\frac{3\gamma P_{b}}{R_{b}}\frac{dR_{b}}{dt}-\frac{6(\gamma -1)k_{l}(T_{bl}-T_{\infty })}{\delta R_{b}}$$
where *γ* is the specific heat ratio of water. The thickness of the thermal boundary layer, *δ* can be calculated by assuming that the profile of the thermal boundary layer adjacent to the bubble wall is quadratic [60]. Since the ideal gas law applies at the bubble center, the gas temperature over time at the bubble center can be written as follows:(28)$$\frac{dT_{bo}}{dt}=-\frac{3(\gamma -1)T_{bo}}{R_{b}}\frac{dR_{b}}{dt}-\frac{6(\gamma -1)k_{l}T_{bo}(T_{bl}-T_{\infty })}{\delta R_{b}P_{b}}$$
where *T_bo_* and *T_bl_* are the temperature at the bubble center and bubble wall, respectively, and k_l_ is the heat conductivity of liquid. Finally, we can derive the time-dependent thickness of the thermal boundary layer from the mass and energy equations for the liquid adjacent to the bubble wall using the integration method:(29)$$\begin{array}{c} \left[1+\frac{\delta }{R_{b}}+\frac{3}{10}{\left(\frac{\delta }{R_{b}}\right)}^{2}\right]\frac{d\delta }{dt}=\frac{6\alpha_{l}}{\delta }-\left[\frac{2\delta }{R_{b}}+\frac{1}{2}{\left(\frac{\delta }{R_{b}}\right)}^{2}\right]\frac{dR_{b}}{dt}\\ -\delta \left[1+\frac{\delta }{2R_{b}}+\frac{1}{10}{\left(\frac{\delta }{R_{b}}\right)}^{2}\right]\frac{1}{\left(T_{bl}-T_{\infty }\right)}\frac{dT_{bl}}{dt} \end{array}$$
where *α*_l_ is the heat diffusivity of the liquid. The temperature at the bubble wall can be derived by solving the energy equation for the gas inside the bubble:(30)$$T_{bl}=\left(T_{\infty }+\frac{k_{g}}{k_{l}}\frac{\delta }{R_{b}}T_{bo}\right)/\left(1+\frac{k_{g}}{k_{l}}\frac{\delta }{R_{b}}\right)$$
where *k_g_* is the heat conductivity of the gas inside the bubble.

The far-field pressure from the evolving bubble can be derived by assuming that the bubble is a monopole source [61]:(31)$$p_{far}=\frac{\rho_{\infty }{\ddot{V}}_{b}}{4\pi r_{d}}=\frac{\rho_{\infty }}{r_{d}}\left(R_{b}^{2}{\ddot{R}}_{b}+2R_{b}{\dot{R}}_{b}^{2}\right)$$Equation (31) can be re-written at the initial point as
(32)$$p_{far}=\frac{R_{o}}{r_{d}}\left(P_{o}-P_{f}\right)$$

## 5. Explosive Boiling on Ultra-Thin Platinum Wire

Skripov and Pavlov [26] measured a superheat limit of 575.2 K for water using a 106 °C/s heating rate on a 20 μm diameter platinum (Pt) wire in a 20 °C water pool. In their case, the nucleation was estimated to be approximately 10^19.5^/(cm^3^s). Glod et al. [28] studied the explosive boiling of water on a thin Pt wire with a diameter of 10 μm. The change in resistivity of the Pt wire was used to determine the temperature of the wire. Using the same procedure as Avedisian et al. [33], the nucleation onset temperature was measured at the temperature curve’s inflection point. Glod et al. also measured the pressure pulse produced by an exploding bubble. They calculated the maximum superheat limit of water, 576.2 K, when the wire’s heating rate exceeded 60 × 10^6^ K/s. At a heating rate of 86 × 10^6^ K/s, the Pt wire surface was seen to be instantly covered with a thin vapor film.

## 6. Results and Discussion

Table 1 shows the physical properties of water, including density, surface tension, saturation pressure, and number density at different nucleation temperatures as measured by Glod et al. [28]. The nucleation rate calculated using the classical nucleation theory at these temperatures is also shown in Table 1. Using classical nucleation theory, the nucleation rate is approximately 3.3 × 10^4^ bubbles/(cm^3^s) (=3.3 × 10^−2^ bubbles/(cm^3^/μs), which shows that no bubble nucleation occurs for microseconds; however, hundreds of bubbles with a volume of cubic cm per second are formed. According to classical nucleation theory, bubble formation does not occur below 566 K. Vargaftik et al. [62] provided the values of surface tension at various temperatures of water, which is an important parameter in classical nucleation theory. The following equation for the temperature dependence of the surface tension of water [62] was used to calculate the temperature dependence of the nucleation rate in classical nucleation theory:(33)$$\sigma =235.8\times 10^{-3}{\left(1-T_{r}\right)}^{1.256}\times \left[1-0.625(1-Tr)\right]$$The unit of surface tension in Equation (33) is N/m, and *T_r_* is reduced temperature.

The radius of the critical-size bubble estimated by Equation (1) is nm at the nucleation temperatures shown in Table 1. It is questionable whether macroscopic surface tension is valid for nanoscale bubbles. Kwon et al. [63] measured the curvature dependence of the surface tension of water in a capillary-condensed water nano-meniscus using a hybrid-force measurement system that combines a tapping-mode, amplitude-modulation atomic force microscope and a microelectromechanical system. The measured surface tension of water at 25 ℃ is approximately 9.59 mN/m for a meniscus radius of curvature of 19 nm.

**Table 1 entropy-26-00035-t001:** Properties of water at several nucleation temperatures.

Nucleation Temperature (K)	548.2 K (*T_r_* = 0.847)	566.2 K (*T_r_* = 0.875)	576.2 K (*T_r_* = 0.890)
Density (kg/m^3^)	759.0	726.0	706.0
Surface tension (N/m)	0.0201	0.0159	0.0136
Saturation pressure (bar)	59.42	77.76	89.49
Number density (molecules/(μm)^3^)	2.539 × 10^10^	2.428 × 10^10^	2.360 × 10^10^
Critical radius (nm), Equation (1)	6.89	4.15	3.08
Nucleation rate by classical nucleation theory (bubbles/cm^3^s), Equation (4)	7.7 × 10^−196^	1.65 × 10^−30^	3.8 × 10^4^

When the molecular cluster model was used, the results were completely different. Table 2 shows the calculation results for the nucleation rate, number of molecules in the critical cluster, and time delay for nucleation at 576.2 K. Table 2 also shows the results of these calculations for T = 566.2 K. The nucleation rate is approximately 2.2 × 10^11^ clusters/(μm^3^μs) at T = 576.2 K and 0.83 × 10^11^ clusters/(μm^3^μs) at T = 566.2 K.

Allowing for the dt increment of the evaporated liquid layer, *dl_L_*, the total number of evaporated molecules in the volume *πD*^2^ *dl_L_* is
(34)$$\pi D^{2}dl_{L}N$$Because the evaporation rate, or the number of molecules evaporated per unit time and unit volume, equals *J_nc_n_c_,* then the number of molecules in a unit volume during *dt* is
(35)$$J_{vol}n_{c}\cdot \pi D^{2}dt$$By equating Equations (32) to (33), one may obtain the evaporation speed of the liquid layer using
(36)$$\frac{dl_{L}}{dt}=\frac{J_{nc}n_{c}}{N}$$For surface nucleation, the evaporation speed can be obtained from the following equation:(37)$$\frac{dl_{L}}{dt}=\frac{J_{sur}n_{c}}{N^{2/3}}$$The evaporation speed obtained by Equation (36) at T = 576.2 K is approximately 116.0 m/s, which is eight times faster than the evaporation speed of butane at the superheat limit. At T = 566.2 K, the evaporation speed is approximately 88.0 m/s. The extremely high nucleation rate and evaporation speed values in both situations are sufficient to produce explosive boiling, a kind of rapid phase change. The nucleation process has a time delay of less than a microsecond, indicating that nucleation occurs immediately after the liquid reaches its superheat limit.

The volume of liquid involved in the nucleation process can also be estimated as follows. When the heater surface reaches a specific temperature, the heat spreads into the liquid layer adjacent to the surface. The following equation can be used to calculate the thickness of the heating layer where evaporation occurs.
(38)$$\sqrt{\alpha_{f}t_{nucl}}$$
where *t_nucl_* is the time duration of nucleation. Using Equation (38), the evaporation thickness is approximately 0.15 μm when the thermal diffusivity is 0.15 × 10^6^ (μm)^2^/s and the nucleation time is 0.15 μs. Furthermore, assuming that the nucleation area of the Pt wire is *πD*^2^ (where *D* is the diameter of the Pt wire), the nucleation volume is approximately 50 (μm)^3^.

The measured nucleation temperature of the Pt wire is 548.15 K [28] at a low heating rate of 10^5^ K/s. Table 3 shows the results for nucleation rate, number of molecules within the critical cluster, nucleation time delay, number of molecules involved in the nucleation process, and evaporation speed. When compared to the results for hydrocarbons at the superheat limit, calculations using the molecular interaction model show that homogeneous nucleation of water is possible at 548.15 K. However, the nucleation process is expected to be slower than in the case of T = 576.2 K. It can be concluded that homogeneous nucleation of water in the micro-Pt wire occurs in a reduced temperature range between 0.85 and 0.89, depending on heating rates. Similar results were obtained for the nucleation of methanol, ethanol, butanol, and heptane in platinum films [35]. Depending on the pulse heating rates, the nucleation of those liquids occurs over a reduced temperature range between 0.83 and 0.91.

The nucleation process of water on the Pt wire is a surface phenomenon; the nucleation rate can be calculated using Equation (24). Table 3 also includes the calculation results for the nucleation rate, number of molecules in the critical cluster, evaporation speed of the liquid layer, and nucleation time delay when the Pt–water contact angle is 60°. Although the assigned nucleation rates are significantly different, similar results are obtained for the number of molecules within a cluster, the evaporation speed, and the nucleation time delay when compared to the results obtained for volume nucleation using Equation (13).

Figure 2 shows the boiling superheat of water on a smooth surface as a function of liquid-side contact angle for nucleation rates of 10^13^ and 10^21^ nuclei/cm^2^s in the molecular cluster model and for a nucleation rate of 10^13^ nuclei/cm^2^s in the classical nucleation theory. After 50° of contact angle, the boiling superheat decreases. In both cases, as the contact angle approaches 180°, the boiling superheat of water approaches 100 °C, the boiling point of water. The experimentally obtained boiling superheat of water at a contact angle of 60°, which is the contact angle of water on the platinum surface, was 556 K [33], which is similar to the superheat calculated with a nucleation rate of 10^13^ clusters/(cm^2^s), as shown in Figure 2. However, the superheat limit obtained by classical nucleation theory with a nucleation rate of 10^13^/(cm^2^s) is about 583.8 K. The boiling superheat of water on the Teflon AF surface was measured to be approximately 9 K above the boiling point (382.2 K) [57]. The Teflon AF surface’s water contact angle may be approximately 167.5° using the results in Figure 2.

Ching et al. [35] tested bubble nucleation of several organic liquids on stressed-minimized platinum films of sizes 4 × 60 μm and 4 × 80 μm by pulse heating of different durations. Table 4 shows that the molecular interaction model predicts the probability of bubble nucleation at different temperatures at which nucleation was observed. The highest nucleation temperature associated with the highest heating rate can be predicted with a nucleation rate of 7 × 10^17^ clusters/(cm^2^s), which is close to the value of 10^17^/(cm^2^s) estimated by Ching et al. In this calculation, the contact angle on the liquid side is assumed to be 60°.

For the boiling at 566.2 K, bubble oscillation after nucleation was calculated using Equations (26)–(29) with the assumption that the bubble formed on the Pt wire is spherical. The Runge–Kutta numerical method was used for the calculations with appropriate initial conditions such as Ro = 6.6 μm, U_o_ = 0, P_o_ = 77.7 bar, T_o_ = 566.2 K, δ_o_ = 0.3 R_o_, and T_bl_ = 350 K. The pressure inside the bubble was obtained by using Equation (11) as one of the initial conditions. The bubble radius was obtained by assuming that all molecules in the nucleated liquid volume form one bubble, and that the pressure inside the bubble is equivalent to the vapor pressure at the nucleation temperature. The bubble’s initial state can be considered an evolved state from the evaporation state that holds the liquid volume. Equation (31) was used to calculate the far-field pressure signal from this evolving bubble at a distance of r_d_ = 20 mm.

The time rates of change of bubble radius and pressure inside the bubble, shown in Figure 3, are very similar to those obtained for droplet evaporation at the superheat limit [58]. Figure 4 shows the far-field pressure signal obtained from the oscillating bubble in Figure 3. Although the maximum magnitude and oscillation period of the far-field signal obtained for the bubble are similar to the experimental data, the temporal rate of change of the far-field pressure signal is significantly different from the experimental results [28]. This is because the behavior of bubbles formed on Pt wires differs significantly from that of spherical bubbles.

## 7. Conclusions

Various experimental methods and techniques confirmed that liquids evaporate at their superheat limit. Herein, bubble nucleation initiated by the evaporation of liquid water on micro-Pt wires was investigated using a molecular interaction model for bubble nucleation. The nucleation rate estimated by the molecular interaction model was approximately 2.1 × 10^11^ clusters/(μm^3^μs) at 576.2 K, which is the superheat limit of water. At 576.2 K, the liquid layer on the Pt surface evaporates at a rate of 116 m/s, which is eight times faster than the evaporation rate of a butane droplet at the superheat limit [17,18]. The nucleation rate (or evaporation speed) is substantially high, and the time delay of the nucleation process is on the order of nanoseconds. This indicated that explosive boiling or rapid phase transition occurs at the superheat limit of water, as observed in experiments. Assuming that the surface occurred on the surface of the micro-Pt wire, similar results were obtained for nucleation, including the number of molecules inside the critical cluster, nucleation time delay, and evaporation rate of the liquid layer. Homogeneous nucleation of water in the micro-Pt wire can occur at a reduced temperature range between 0.85 and 0.89, depending on the heating rate. Similar results obtained for the nucleation temperatures of methanol, ethanol, butanol, and heptane on stress-minimized platinum film heaters [35] can be predicted by the molecular interaction model. The probability of water nucleation occurring on the Pt wire surface at 576.2 K over a period of several seconds and over a space as large as cubic cm was predicted using classical bubble nucleation theory. The behavior of bubbles formed on the micro-Pt wire could not be explained using the behavior of spherical bubbles.

## Figures and Tables

**Figure 1 entropy-26-00035-f001:**
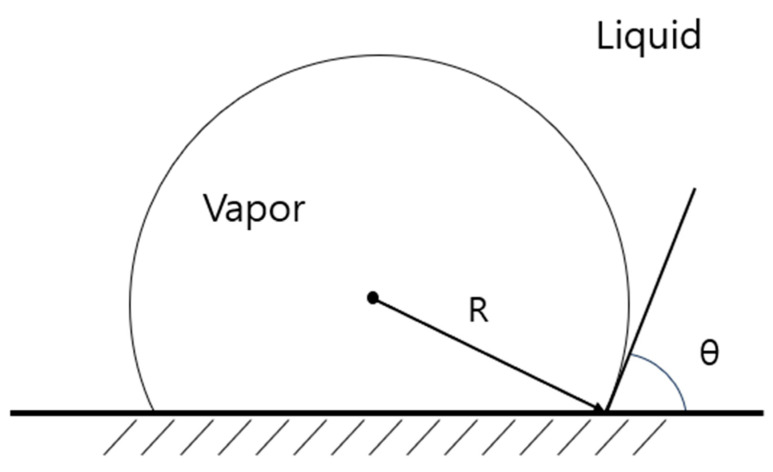
A bubble formed on a smooth surface at the atomic scale with a liquid-side contact angle of θ.

**Figure 2 entropy-26-00035-f002:**
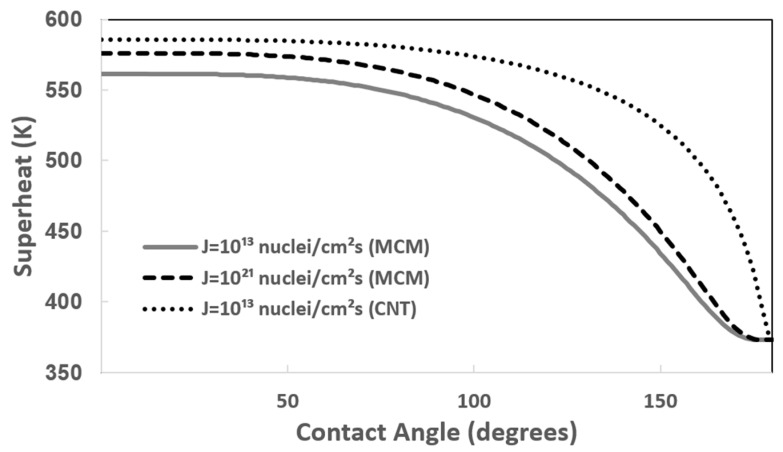
Boiling superheat temperature (K) of water depending on liquid-side contact angle with nucleation rates of *J_sur_* = 10^13^ (solid line) and *J_sur_* = 10^21^ nuclei/(cm^2^s) (slash line) in the molecular interaction model and with nucleation rate of *J_CNT_*,*_sur_* = 10^13^ (dotted line) in the classical nucleation theory. MCM and CNT stand for molecular cluster model and classical nucleation theory, respectively.

**Figure 3 entropy-26-00035-f003:**
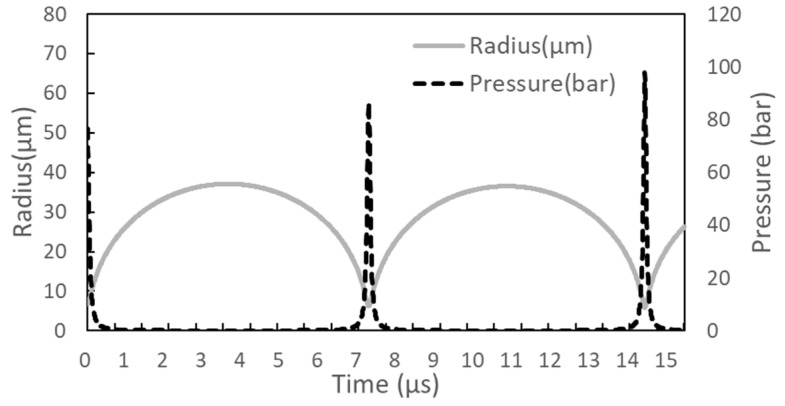
Time rate change of bubble radius and the pressure inside the bubble, whose initial state is Ro = 6.6 μm and P_o_ = 77.7 bar.

**Figure 4 entropy-26-00035-f004:**
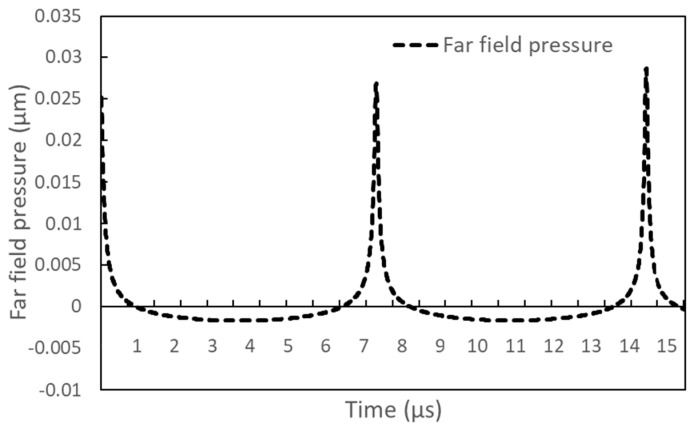
Pressure wave signal from the oscillating bubble formed on the Pt wire.

**Table 2 entropy-26-00035-t002:** Various nucleation parameters obtained by the molecular interaction model at nucleation temperatures of 576.2 K and 566.2 K.

Nucleation Temperature (K)	566.2 K	576.2 K
Nucleation rate (clusters/(μm^3^μs))	0.83 × 10^11^	2.15 × 10^11^
Number of molecules in the critical cluster	25.75	12.74
Evaporation speed of the liquid layer (m/s)	88.0	116.0
Total number of molecules involved in the nucleation process	1.21 × 10^12^	1.27 × 10^12^
Time lag for nucleation (μs)	0.0184	0.00572
Pressure inside bubble (bar)	77.7	89.5
Pressure of evaporated state (bar)	905.0	827.0

**Table 3 entropy-26-00035-t003:** Various nucleation parameters obtained by the molecular interaction model at a nucleation temperature of 548.2 K.

Nucleation Process	Equation (13)	Equation (24)
Nucleation rate (clusters/(μm^3^μs))	4.26 × 10^9^ (clusters/(μm^3^μs))	3.16 × 10^6^ (clusters/(μm^2^μs))
Number of molecules in the critical cluster	88.4	74.6
Evaporation speed of the liquid layer (m/s)	14.8	27.3
Total number of molecules involved in the nucleation process	1.2 × 10^12^	1.2 × 10^12^
Time lag for nucleation (μs)	0.142	0.422
Pressure inside bubble (bar)	59.41	59.41
Pressure of evaporated state (bar)	1040	1040

**Table 4 entropy-26-00035-t004:** Properties of heptane at several nucleation temperatures and nucleation rate by molecular interaction model and classical bubble nucleation theory.

Nucleation Temperature (K)	449.4 K (*T_r_* = 0.832)	472.0 K (*T_r_* = 0.874)	484.3 K (*T_r_* = 0.897)
Density (kg/m^3^)	527.0	495.0	476.0
Surface tension (N/m)	0.0059	0.0042	0.0033
Saturation pressure (bar)	6.19	9.45	11.84
Nucleation rate by molecular interaction model (clusters/cm^2^s)	1.6 × 10^15^	2.3 × 10^17^	6.9 × 10^17^
Nucleation rate by classical nucleation theory (bubbles/cm^2^s)	0.0	2.3 × 10^−73^	1.6 × 10^−3^

## Data Availability

Data generated during the study are unavailable due to ethical restrictions.

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
