# Peer review of "Nucleation Process in Explosive Boiling Phenomena of Water on Micro-Platinum Wire"

_entropy, 2023, doi:10.3390/e26010035_

Round 1

Reviewer 1 Report (Previous Reviewer 2)

Comments and Suggestions for Authors

The authors made several significant improvements to the manuscript, making it easier to read. Especially the extensive nomenclature helps. It is now also clear to me that they apply CNT correctly. I detected no obvious errors.

Comments on the Quality of English Language

I still don't think the exposition and the language are good. One has to read between the lines to understand what is meant. To take a concrete example, consider the first sentence in the conclusions:

"The evaporation process of liquid occurs at the superheat limit of the liquid, and this has been confirmed through various experimental methods and techniques."

The evaporation process of a liquid usually occurs at very different conditions than at the superheat limit, so in that sense I would say their sentence is misleading/confusing. Sure, they study the evaporation process at the superheat limit in this paper, but these sentence constructions add a lot of mental friction for the reader.

For this reason, I think the impact of this paper will be limited. On the other hand, the current version is much better than the original draft. Since there are no clear errors I won't insist that it be rejected. I'll leave that decision with the editor.

Author Response

Replies to Reviewer 1’s Comments:

The authors made several significant improvements to the manuscript, making it easier to read. Especially the extensive nomenclature helps. It is now also clear to me that they apply CNT correctly. I detected no obvious errors.

Comment 1: Comments on the Quality of English Language

I still don't think the exposition and the language are good. One has to read between the lines to understand what is meant. To take a concrete example, consider the first sentence in the conclusions:

"The evaporation process of liquid occurs at the superheat limit of the liquid, and this has been confirmed through various experimental methods and techniques."

The evaporation process of a liquid usually occurs at very different conditions than at the superheat limit, so in that sense I would say their sentence is misleading/confusing. Sure, they study the evaporation process at the superheat limit in this paper, but these sentence constructions add a lot of mental friction for the reader.

Reply to comment 1: This sentence is intended for the case of superheat limit, not other evaporation processes. Anyway, this sentence was rewritten in the revised manuscript. The English of our manuscript has been proofread by a professional English editor.

For this reason, I think the impact of this paper will be limited. On the other hand, the current version is much better than the original draft. Since there are no clear errors I won't insist that it be rejected. I'll leave that decision with the editor.

Reviewer 2 Report (New Reviewer)

Comments and Suggestions for Authors

Summary: This manuscript uses a molecular interaction model to study phase change nucleation processes in micro-platinum wires. The study presents results indicating that the nucleation statistics derived from the molecular interaction model differ from those initially observed in classical nucleation theory models. The authors demonstrate that contrary to classical nucleation theory, bubbles not only emerge at temperatures below 566K but also display explosive boiling behavior. The study further reports elevated nucleation rates and liquid evaporation speeds at different temperatures as well. The manuscript potentially provides new findings related to homogeneous phase-change heat transfer, but some questions must be adequately answered.

Major:

1.    The authors discuss classical nucleation theory, which pertains to homogeneous nucleation, while practical boiling processes often align more closely with heterogeneous nucleation theory. Even the reference (Glod et al) cited by the authors for property evaluation utilizes empirical correlations to estimate nucleation rates, accounting for heterogeneous nucleation. It is essential for the authors to clarify the rationale behind selecting homogeneous nucleation models as their reference model.

2.    Consistent with the initial comment, the anticipated nucleation temperatures needed for homogeneous nucleation are typically considerably higher than those necessary for initiating boiling.[1] Therefore, the assertion in this article that bubbles are not expected to form at temperatures below 566K has been documented in references for many decades. If the mechanism diverges from heterogeneous nucleation theory, please provide a detailed explanation in the manuscript.

3.    The conclusion of the manuscript is vague. Please provide clear and concise summaries, contributions, and conclusions of the work for clarity.

Minor:

1.    Kindly standardize the units (e.g., bubbles/cm3s or bubbles/μm3μs) consistently in both the tables and manuscript. The need to convert back and forth was confusing and hindered a clear understanding of the message the authors intended to convey.

2.    It is recommended to maintain consistency in the order of temperature in tables. If you commence with lower temperatures and progress to higher ones, please maintain this sequence consistently across all tables to enhance reader comprehension.

3.    Line 327-328. Do you mean at 576.2K? If so, why is it different from the values shown in Table 1?

Reference

[1] Rohsenow, Warren M., James P. Hartnett, and Young I. Cho. Handbook of heat transfer. Vol. 3. New York: Mcgraw-hill, 1998.

Comments on the Quality of English Language

Kindly clarify the authors' contributions and elucidate the study's findings and implications.

Author Response

Replies to Reviewer 2’s Comments

Summary: This manuscript uses a molecular interaction model to study phase change nucleation processes in micro-platinum wires. The study presents results indicating that the nucleation statistics derived from the molecular interaction model differ from those initially observed in classical nucleation theory models. The authors demonstrate that contrary to classical nucleation theory, bubbles not only emerge at temperatures below 566K but also display explosive boiling behavior. The study further reports elevated nucleation rates and liquid evaporation speeds at different temperatures as well. The manuscript potentially provides new findings related to homogeneous phase-change heat transfer, but some questions must be adequately answered.

Major:

Comment 1:   The authors discuss classical nucleation theory, which pertains to homogeneous nucleation, while practical boiling processes often align more closely with heterogeneous nucleation theory. Even the reference (Glod et al) cited by the authors for property evaluation utilizes empirical correlations to estimate nucleation rates, accounting for heterogeneous nucleation. It is essential for the authors to clarify the rationale behind selecting homogeneous nucleation models as their reference model.

Reply comment 1: Research on nuclear boiling began in the 50s and was actively conducted in the 60s and 70s. Nucleate boiling is characterized by a low superheat, approximately 1 to 10 K above the boiling point of the liquid. Clark et al. [2] observed with high-speed cameras that nucleate boiling sites identified as pits on the surface. Cornwell [3] identified natural sites where nucleate boiling occurs on copper surface by scanning electron microscope (SEM). Nail et al. [4] identified the nucleate boiling point via SEM as a cavity with a radius ranging from 0.4 to 5 μm. Nucleate boiling is characterized by the departure frequency of bubble and number of nucleation sites, which is related to boiling heat transfer [5]. Trefethen [6] showed that a liquid can overheat when it is surrounded by another liquid. Inspired by Trefethen's observations, Wakeshima and Takata [7], and a year later Moore [8], performed droplet explosion experiments to obtain the theoretical superheat limit [9] that could be addressed by homogeneous nucleation. This is why the homogeneous nucleation model applies to boiling on smooth surfaces at the atomic scale without cavities [10, 11]. In the revised manuscript, the above discussion was addressed from L 28 to L 41.

Comment 2: Consistent with the initial comment, the anticipated nucleation temperatures needed for homogeneous nucleation are typically considerably higher than those necessary for initiating boiling.[1] Therefore, the assertion in this article that bubbles are not expected to form at temperatures below 566K has been documented in references for many decades. If the mechanism diverges from heterogeneous nucleation theory, please provide a detailed explanation in the manuscript.

Reply to comment 2: The boiling phenomenon discussed in Reference 1 is nucleate boiling, which is initiated by surface cavities. By eliminating these surface cavities, homogeneous bubble nucleation can be achieved. In the 90s, IC-based microfabrication techniques made it possible to create atomically smooth surfaces without cavities. Boiling on the smooth surfaces in atomic scale occurred at very large superheats near the critical temperature of the liquid [12]. Boiling on smooth surfaces in atomic scale occurs very rapidly and is characterized by microseconds, which cannot be handled by the nucleate boiling phenomena. Homogeneous nucleation of water on micro-Pt wire [13] and homogeneous nucleation of methanol, ethanol, butanol, and heptane on platinum film occur in a reduced temperature range between 0.83 to 0.91 [14]. This argument was discussed from L 411 to L416.

A detailed description of this argument is given in paragraphs L28 through L41.

Comment 3:    The conclusion of the manuscript is vague. Please provide clear and concise summaries, contributions, and conclusions of the work for clarity.

Reply to comment 3: Conclusion was rewritten based on the reviewer’s suggestions.

Minor:

Comment 1:    Kindly standardize the units (e.g., bubbles/cm3s or bubbles/μm3μs) consistently in both the tables and manuscript. The need to convert back and forth was confusing and hindered a clear understanding of the message the authors intended to convey.

Reply comment 1: It is intended to help readers understand the context of nucleation at a given time and space. If nucleation occurs in micro-dimensions over microseconds, the nucleation unit of bubbles/μm3μs is a better way to understand the situation.

Comment 2:    It is recommended to maintain consistency in the order of temperature in tables. If you commence with lower temperatures and progress to higher ones, please maintain this sequence consistently across all tables to enhance reader comprehension.

Reply to comment 2: According to the reviewer's suggestion, the temperature order was changed from low to high in the revised manuscript. See Table 2.

Comment 3:    Line 327-328. Do you mean at 576.2K? If so, why is it different from the values shown in Table 1?

Reply comment 3: Yes, in terms of classical nucleation theory, this is the case at 576.2K. Avoiding confusion, the sentence was rewritten. In the revised manuscript, issues related to the macroscopic surface tension values used in classical nucleation theory were discussed from L 353 to 359.

References

[1] Rohsenow, Warren M., James P. Hartnett, and Young I. Cho. Handbook of heat transfer. Vol. 3. New York: Mcgraw-hill, 1998.

[2] Clark, H. B., Strenge, P.S., Westwater, J.W. Active sites for nucleate boiling, CEP Symp. Ser. 29, vol. 55, pp. 103- , 1959.

[3] Cornwell, K. Naturally formed boiling site cavities, Lett. Heat Mass Transfer, vol. 4, pp. 63-762, 1977.

[4] Nail, J. P., Vachon, R. I., Morehouse, J. An SEM study of nucleation sites in pool boiling from 304 stainless steel, Transactions of ASME, J. Heat Transfer, Vol. C96, pp. 132-137, 1974.

[5] Bergles, A. E., Rohsenow, W. M. The determination of forced-convection surface-boiling heat transfer, Transactions of ASME, J. Heat Transfer, vol. C86, pp. 365-372, 1964.

[6] Trefethen, L. Nucleation at a liquid-liquid interface. J. Appl. Phys. 1957, 28, 923-924.

[7] Wakeshima, H.; Takata, K. On the limit of superheat. J. Phys. Soc. Jap. 1958, 13, 1398-1403.

[8] Moore, G. R. Vaporization of superheated liquids. AIChE J. 1959, 5, 458-466.

[9] Döering, W. Die Überhitzungsgrenze und Zerreiβfestigkeit von Flϋssigkeiten. Z. Phys. Chem. 1937, 36, 1398-1403.

[10] Avedisian, C.T.; Osborne, W. S.; McLeod, F. D.; Curley, C. M. Measuring bubble nucleation temperature on the surface of a rapidly heated thermal ink-jet heater immersed in a pool of water. Proc. R. Soc. Lond. A, 1999, 455, 3875-3899.

[11]  Lee, J. Y.; Park, H.C.; Jung, J. Y.; Kwak, H. Bubble nucleation on micro line heater. ASME  J. Heat. Trans. 2003, 125, 687-692.

[12] Lin, L.; Pisano, A. P. Thermal bubble powered micro actuators, Microsystem Tech., vol. 1, pp. 51-58, 1994.

[13] Glod, S.; Poulikakoa, D; Zhao, Z; Yadigaroglu, G. An investigation of microscale explosive vaporization of water on an ultrathin Pt wire. Int. J. Heat Mass Trans. 2002, 45, 367-379.

[14] Ching, E. J.; Avedisian, C. T.; Cavicchi, R. C.; Chung, D. H.; Rah, K. J.; Carrier, M. J. Rapid evaporation at the superheat limit of methanol, ethanol, butanol and n-heptane on platinum films supported by low stress SiN membranes. Int. J. Heat Mass Trans. 2016, 101, 707-718.

Reviewer 3 Report (New Reviewer)

Comments and Suggestions for Authors

It is very difficult to identify the contribution of the authors to this work. Most of the writing, if not all, is referring to others' works.  Novelty is lacking. Please see the attached document for other comments.

Comments on the Quality of English Language

Clarity is lacking.

Round 2

Reviewer 2 Report (New Reviewer)

Comments and Suggestions for Authors

Comment 1:

The provided response falls short of adequately addressing my inquiry, instead delving extensively into historical context. As stated by the author, the studies referenced (Wakeshima, Takata, and Moore) focus exclusively on droplet explosion experiments rather than nucleate boiling on surfaces. This is concerning as these two experiments are inherently different. Although the concept of homogeneous nucleation on atomic scale surfaces is theoretically plausible, I am not aware of any experimental studies substantiating this claim. The citation the author relies on to support the occurrence of homogeneous bubble nucleation on smooth surfaces (reference 12, Blander et al) pertains, in reality, to another droplet explosion experiment.

I respectfully request recent references that definitively demonstrate the possibility of homogeneous boiling on smooth surfaces. Additionally, given the substantial challenges associated with achieving atomic scale surfaces in practical applications, I am interested in receiving further elucidation on the justification for these systems and insights into their potential real-world applications.

Comment 2:

It appears that the reference cited by the author to assert the achievability of homogeneous nucleation on micro-PT wires is, upon closer examination, another instance of a droplet explosion experiment involving resistive wires wrapped around the vessel for heating. I request a thorough verification of this information and the provision of more relevant references that explicitly demonstrate homogeneous nucleation occurring on boiling surfaces.

Comment 3:

In the conclusion, please ensure greater precision in language. This study did not conduct experiments; instead, it tested another theory based on past data points.

Author Response

Reviewer 3 Report (New Reviewer)

Comments and Suggestions for Authors

At the beginning of the introduction, it should 'boiling on solid surface' not "boiling of the solid surface".

Comments on the Quality of English Language

N/A

Author Response

//

Comment 1: At the beginning of the introduction, it should 'boiling on solid surface' not "boiling of the solid surface".

Reply to Comment 1: It was corrected.

Round 3

Reviewer 2 Report (New Reviewer)

Comments and Suggestions for Authors

The authors have adequately addressed my inquiries, and thus, I recommend this article for publication.

This manuscript is a resubmission of an earlier submission. The following is a list of the peer review reports and author responses from that submission.

Round 1

Reviewer 1 Report

Comments and Suggestions for Authors

Comments on the Quality of English Language

The text needs proofreading and correction to avoid typos and grammatical mistakes.

Reviewer 2 Report

Comments and Suggestions for Authors

The topic is somewhat interesting, but the exposition is poor.

I’m not convinced that the authors have applied Classical Nucleation Theory correctly. The superheat limit of liquid water, when applied correctly, is in fact in excellent agreement with experiments (about 575 K), and this has been shown many times in the literature. The superheat limit should not be very sensitive to the specification of nucleation rate, since the rate increases extremely fast with temperature. This would become clearer if the authors plot the superheat limit in a temperature-pressure diagram, e.g. defined by J_CNT = 10^13 m^-3 s^-1.

I’ve never seen the “molecular interaction model” before, but it seems to be very different from what I expect based on classical nucleation theory. In Table 2, it is shown that the nucleation rate increases by a factor less than 3 when increasing the superheat by 10 K; however, it is expected to increase by a lot, lot more. Classical nucleation theory is the established model in the literature, and it should be commented on why the molecular interaction model make extremely different predictions.

Also: where does Eq. 23 come from? What do J_nc and n_c mean in Eq. 23? 

Perhaps there is a worthwhile contribution hidden here, but I’m not able to decipher it. It would have to be written in a much more pedagogical manner for me to give an opinion on that.

General suggestions:

a) On line 100, it should be specified that V_m is the molecular volume of the liquid (not the vapor).

b) In Table (1), please use SI units for surface tension.

c) A nomenclature would ease reading.

Typos:

108: wildly -> widely

115: [31] -> [34]

215: ration -> ratio

Comments on the Quality of English Language

The language is of low quality, and I have difficulties following the reasoning.

Reviewer 3 Report

Comments and Suggestions for Authors

1. The introduction doesn't clearly state the research objectives or questions being addressed.

2. Some statements, particularly regarding previous research, lack proper citations.

3. Key terms like "superheat limit" and "explosive boiling" are introduced without clear definitions.

4. The methods used in the research are mentioned but not thoroughly explained.

5. The study does not compare its findings to existing literature or alternative models.

6. The study does not acknowledge any limitations or potential sources of error.

7. The conclusion is brief and doesn't summarize the key findings or implications of the research.

Comments on the Quality of English Language

Minor editing of English language required.